# Thermal and Principal Ablation Properties of Carbon-Fibre-Reinforced Polymers with Out-of-Plane Fibre Orientation

**Sebastian Eibl *** and **Thomas J. Schuster**

Bundeswehr Research Institute for Materials, Fuels and Lubricants, Institutsweg 1, D-85435 Erding, Germany; thomasschuster@bundeswehr.org
* Correspondence: SebastianEibl@bundeswehr.org

**Abstract:** This work characterises thermal properties of a typical epoxy-based carbon-fibre-reinforced polymer used in aircraft construction, but with an out-of-plane fibre orientation, and assesses its potential as a structural ablative material. Samples of the commercially available Hexply® 8552/IM7 are prepared with out-of-plane angles up to 90°, with a focus on 0° to 15°, enhancing thermal conductivity through the thickness of the panel. Ablation processes are simulated by a hot-air blower at 580 °C, and examined in detail by ultrasonic testing and microfocused computed X-ray tomography afterwards. Matrix degradation is characterised by infrared spectroscopy and mass loss. To assess structural properties, tensile, compression, and bending tests are performed. The results show a loss in mechanical performance with an increasing fibre angle, which may be negligible for angles lower than ~5° in the initial state. Composite material with an out-of-plane fibre orientation is deeply penetrated concerning matrix degradation by thermal loading, but it is held together by the fibres fixed in the intact matrix underneath. This type of material shows a high potential for structural components in single-use, high-temperature, ablative applications with a focus on saving weight.

**Keywords:** carbon-fibre-reinforced polymer composites; fibre orientation; ablation; out-of-plane

## 1. Introduction

Carbon-fibre-reinforced polymer matrix composites (CFRP) are widely used for aerospace, ballistic, and engineering components, etc., because of their low specific weight and excellent mechanical characteristics [1]. As polymers are usually susceptible to thermal damage, many studies deal with their temperature behaviour and the use of flame retardants. For flame retardants for epoxy-based composites [2,3], new developments are typically halogen-free [4–7] and/or nano-scaled [8]. In order to protect CFRP structures from elevated temperatures in, for example, hypersonic applications, ablation processes [9,10] are typically exploited in aeronautical and aerospace construction [11–14]. In an ablation process, high energy inputs are dissipated by the material through endothermic processes, in which the material itself is usually degraded or consumed [15,16]. Depending on the application area, there are different materials for ablation structures and different test methods, such as powered plasma jets, oxy-acetylene torch heaters, rocket exhausts, radiant heating lamps, etc. [17]. Significant properties of an ablative material are density, thermal conductivity, and specific heat capacity, influencing the heat of ablation as well as the heat and mass transfer rate [17,18]. However, additional layers and additives usually result in higher weights. There have already been investigations to enhance the thermal conductivity of lightweight composites, which could be achieved, for example, by conductive fillers such as carbon black and/or carbon nanotubes [19–21]. Park et al. additionally investigated the thermal behaviour of these materials and reported better ablation properties [22]. A simple approach to avoid fillers with their accompanying disadvantages (e.g., higher resin viscosity [23,24]) is to use the carbon fibres themselves to enhance the thermal conductivity and ablation behaviour of CFRP, as their thermal conductivity is much higher

than that of the polymeric matrix. Additionally, carbon fibres resist higher temperatures compared to the polymeric matrix, and during the combustion of CFRP they typically act as barriers concerning heat and mass transfer [25]. A new approach within this work is to change the direction of the fibres out-of-plane to improve thermal conductivity through the thickness of the composite material and to enhance its ablation properties. A fibre orientation with low out-of-plane angles may still provide a protective barrier effect. In addition, fibres at the surface are not as easily ablated as in layered CFRP samples with regular fibre orientation, because these fibres are still locked into deeper positions inside the intact material during ablation. By this, a longer duration is achieved to withstand high temperatures under ablative conditions. However, mechanical properties are supposed to suffer from out-of-plane fibre orientation. Therefore, the mechanical performance of composites with an out-of-plane fibre orientation is analysed in detail before and after the ablation experiments. As this performance is expected to be highly sensitive on fibre orientation and may not be excessively reduced, this study focusses on angles lower than 15°, with prospects up to 90°.

After the ablation experiments, non-destructive evaluation techniques are applied such as ultrasonic testing, microfocused computed X-ray tomography (μCT), and infrared spectroscopy (ATR-FTIR). Infrared spectroscopy provides especially deep insight into degradation processes of the polymer matrix when it is not extensively decomposed.

Samples of the epoxy-based CFRP system Hexply$^{®}$ 8552/IM7 with an out-of-plane fibre orientation used for this work have already been investigated for their reaction-to-fire properties by cone calorimetry [26]. For the presented study, ablation processes are simulated by a hot-air blower at 580 °C. Corresponding tests according to ASTM E285 [27] using an oxy-propane flame are additionally carried out.

The aim of this work is to gain deep insight into degradation processes of CFRP with an out-of-plane fibre orientation at a one-sided thermal load and developing temperatures beyond the decomposition temperature of the polymer matrix. For the first time, their potential is assessed for using them under ablating conditions in structural components with the advantage of not needing additional ablative additives or ablation layers [28].

## 2. Material

The investigated commercially available CFRP system Hexply$^{®}$ 8552/IM7 from Hexcel Composites GmbH consists of an epoxy resin (~29 wt.%) and is toughened with a polyethersulfone (~6 wt.%) [29]. Its maximum operational temperature is 121 °C [30]. A total of 960 prepreg layers are unidirectionally stacked and cured in an autoclave process according to the manufacturer's recommended conditions [30]. No distinguishable fibre layers exist after curing, as carbon fibres are nearly homogeneously distributed throughout the resulting CFRP cube [25]. Test samples 2 mm thick with an out-of-plane fibre orientation are cut out of the CFRP cube in different angles using a water-cooled diamond wire saw, as schematically shown in Figure 1, creating fibre angles between 0° and 90° with respect to the panel surface. All panels contain identical ratios of fibre-matrix content as well as fibre-matrix areas at the surface. Each sample is visually inspected for surface damages and ultrasonically tested to ensure that it is free of voids and delaminations. Samples are subsequently dried in an oven at 70 °C for a minimum of a week.

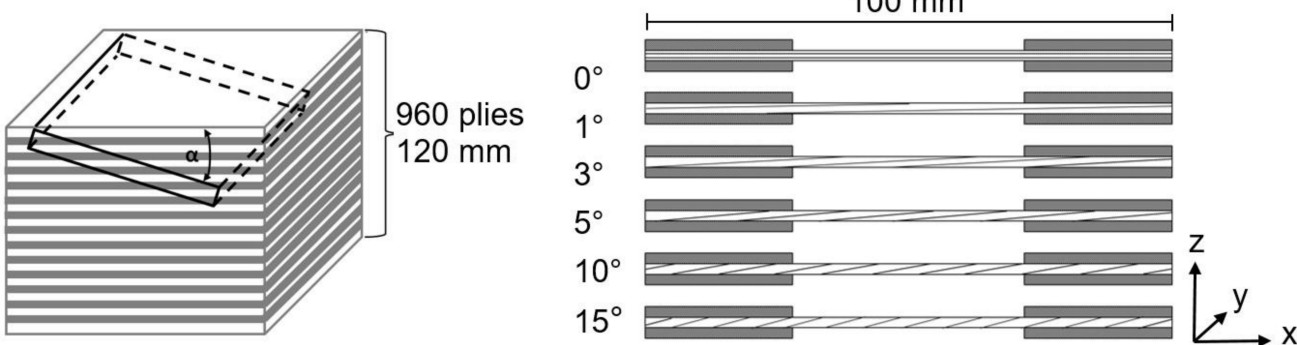

**Figure 1.** Schematic representation of a cube consisting of 960 plies used to cut specimens with out-of-plane fibre angles with respect to the panel surface. Schematic cross-sections of tensile test specimens are shown on the right side.

## 3. Experimental

Thermal loading/ablation tests are predominantly performed using a ~5 kW heat gun ("Lufterhitzer 5000", Leister Type 8D4) using compressed air (~4 bar) with a nozzle diameter of 8 mm and an air temperature at the nozzle of $580 \pm 5$ °C. It is adjusted with a distance of 5 mm to the vertically aligned samples ($20 \times 10 \times 2$ mm$^3$). Samples are clamped in a specially designed metal holder avoiding thermal conduction into the support by four pins. Temperatures are recorded at the free back side of the samples at three positions (see Figure 2) with attached type K thermocouples. Preceding tests using a propane flame for ablation experiments are carried out with a commercial welding gun at an oxygen flow of 374 mL/min and a propane flow of 122 mL/min resulting in a stable flame with an inner cone of 5 mm. For comparison reasons, horizontally aligned samples are irradiated at 60 kW/m$^2$ at a distance of 25 mm from an electrical heater of a cone calorimeter [31] for various durations up to time to ignition.

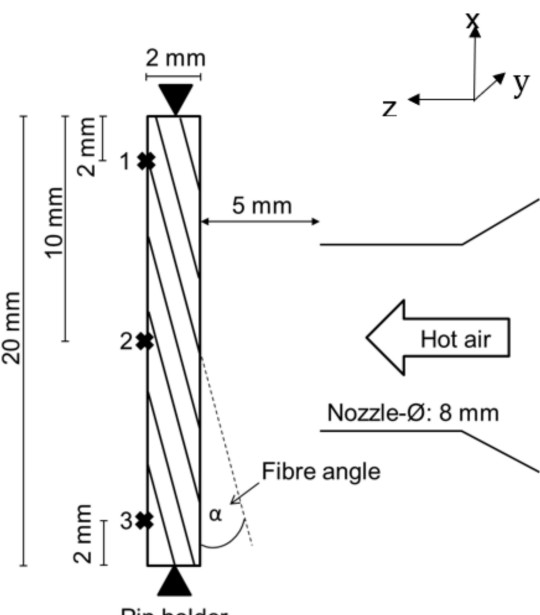

**Figure 2.** Schematic sectional view of the ablation experiment with marked positions for the attached thermocouples and the subsequent FTIR analysis.

Tensile tests are performed according to DIN EN ISO 527-5 [32] using a universal testing machine. Divergent for this standard specification, sample length is reduced to 100 mm due to the size of the CFRP cube. The free sample length is, therefore, 44 mm. Because of these reductions, the sample width is adjusted to 10 mm and thickness to

2 mm. Short beam shear and compressive strength are tested according to DIN EN ISO 14126 [33] and DIN EN ISO 2563 [34], respectively, also using a universal testing machine by ZwickRoell.

Thermogravimetric analysis (TGA) is performed in air using a STA 449 (Netzsch, Selb, Germany) with a heating rate of 10 K/min. Thermal conductivity is calculated as a product of temperature conductivity, density and specific heat capacity, all measured at 15 °C. Measurement of temperature conductivity is performed by a Laser Flash Analysis (LFA) according to DIN EN ISO 22007-4 [35] using a LFA 427 (Netzsch). Heat capacity is measured according to DIN 53765 F [36] on a DSC Q2000 (TA Instruments, New Castle, DE, USA) and density is determined gravimetrically according to DIN EN ISO 845 [37]. Thermal expansion is measured by thermomechanical analysis (TMA), which is carried out with a Netzsch TMA 402 Hyperion system with a heating rate of 3 K/min.

Changes in the composition of the polymer matrix due to the thermal load are analysed by micro Attenuated Total Reflection Fourier Transform Infrared Spectroscopy (µ-ATR-FTIR). Spectra are recorded with a Bruker Tensor 27 spectrometer and a Harrick ATR cell equipped with a silicon crystal (diameter: 0.1 mm) on the specimens' back side at three positions given in Figure 2. Three spectra are recorded for every spot and received band intensities are averaged. Data analysis was performed as described elsewhere [26].

Microfocused computed X-ray tomography (µCT) is performed using a General Electric V-TOME XL300 system with a 300 kV microfocus X-ray source, with a voxel size of 14 µm.

Scanning electron microscopy (SEM) images are recorded at 1.0 kV with a Zeiss Ultra Plus microscope. Ultrasonic scanning is performed according to EN 45000 with a USPC 3010-HFUS 2000 by DR. HILLGER equipped with a Panametrics 20 MHz sensor head with a resolution of 2 µm.

## 4. Results and Discussion

### 4.1. Basic Characterisation of Mechanical and Thermal Properties

The results of the tensile, compression and short beam shear tests for CFRP material with an out-of-plane fibre orientation are shown in Figure 3. As expected, mechanical performance typically decreases with an increasing fibre angle. As for increasing fibre angles, mechanical load is more and more borne by the weaker matrix compared to the fibres. Additionally, failure modes for the various mechanical testing methods change for samples with an out-of-plane fibre orientation. In a tensile test of samples with fibres orientated in load direction (0°), all fibres reach from one clamp to the other, and strength is dominated by fibre properties. For increasing out-of-plane angles, fibres clamped at one side may not reach the other clamp (see Figure 1). Then, mechanical performance is dominated by the matrix and/or fibre/matrix interface properties, as fracture occurs in an interlaminar manner (see Figure 4). Similarly, in short beam shear tests, the failure mode changes from interlaminar for samples with an out-of-plane fibre angle of 0° to cracking along the fibres for samples with a 90° fibre angle. Therefore, fracture areas change with fibre angle and test results are influenced by sample dimensions and experimental setup. Nevertheless, mechanical strength is still calculated on basis of the initial cross-sectional area of the samples for comparison reasons, keeping in mind the mechanistic changes. A discussion of this topic based on laminate theory [38] is beyond the scope of this work, as it focusses on thermal and ablative properties of CFRP material with an out-of-plane fibre orientation. When this material is used for ablative components, a loss in mechanical performance has to be taken into account, which may be, however, tolerable for out-of-plane fibre angels lower than ~5°. For example, a fibre angle of 3° leads to a reduction of 1% for the calculated short beam shear strength (SBSS) using the fixed cross-sections and 24% for tensile and compressive strength compared to the 0° fibre orientation. As expected, the decrease in SBSS is less sensitive to the fibre angle, as the sample dimensions and the experimental setup show less influence on the type of fracture compared to tensile strength. For an out-of-plane fibre orientation of 90°, compressive strength is less reduced

compared to tensile and short beam shear strength, as this strength type is most influenced by matrix properties [39]. Without the influence of the dimensions of the test specimens on mechanical properties, the loss of mechanical performance of a composite component might be less pronounced and compensated with slightly thicker material.

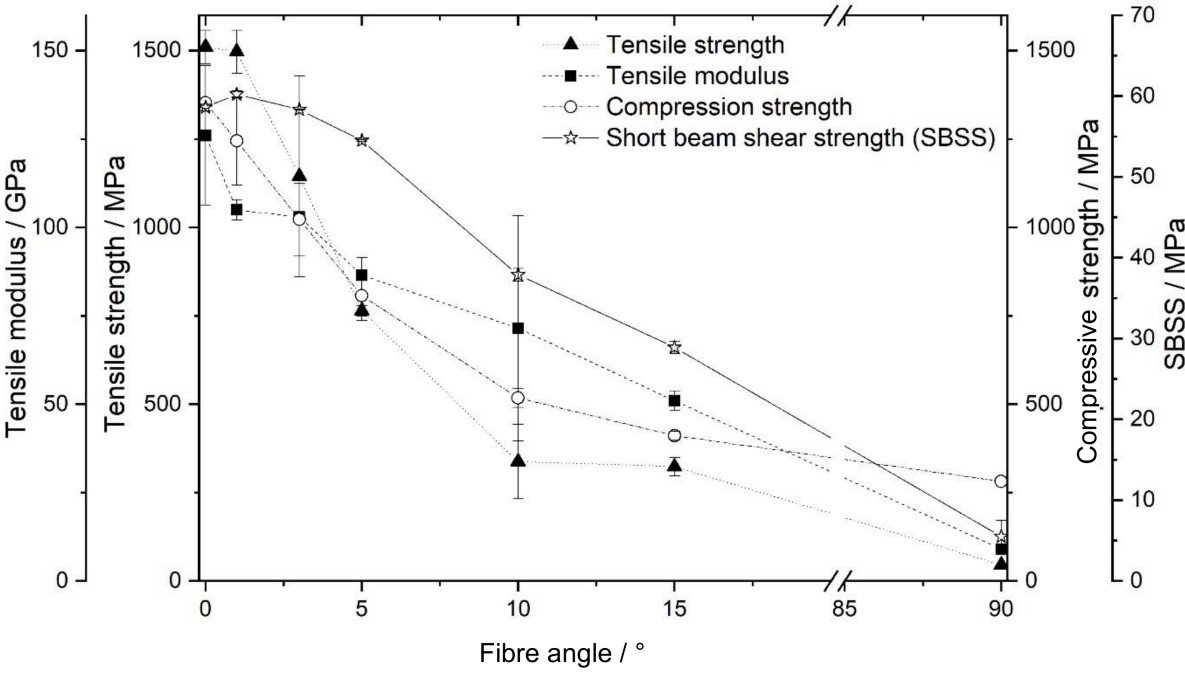

**Figure 3.** Mechanical performance of CFRP samples with an out-of-plane fibre orientation. (Given strengths are calculated for the initial cross-sections of the samples and not the area of failure (see text).)

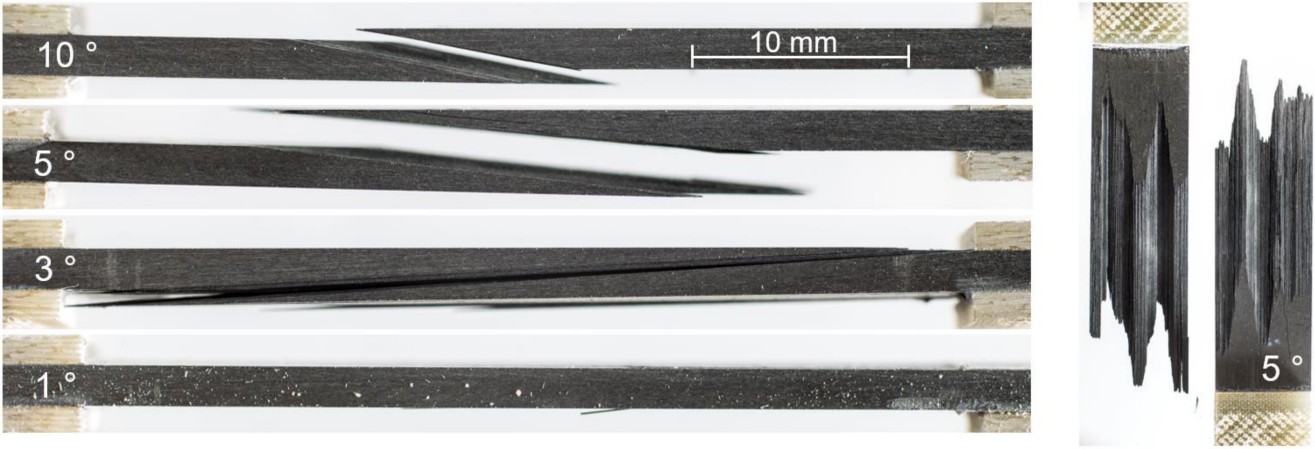

**Figure 4.** Thermally not treated samples with out-of-plane fibre orientation after tensile testing.

An important property for ablative materials is their thermal conductivity. Thermal energy is mainly conducted by the carbon fibres, as they show higher thermal conductivity than the surrounding polymers. Therefore, the through-thickness thermal conductivity of the composite rises significantly with an increasing fibre angle, as shown in Figure 5. The thermal conductivity of composites with a 90° fibre angle reaches the value of the used carbon fibres, which is given as 5.40 W/m·K [40]. The angle dependency of the thermal conductivity (*TC*) may also be simulated by Equation (1), with $TC_{0°}$ being the thermal

conductivity through the thickness of samples with an out-of-plane angle ($\alpha$) of 0° and $TC_{90°}$ for 90°, as described by Hasselman et al. [41]:

$$TC = TC_{0°} \cdot cos^2(\alpha) + TC_{90°} \cdot sin^2(\alpha) \tag{1}$$

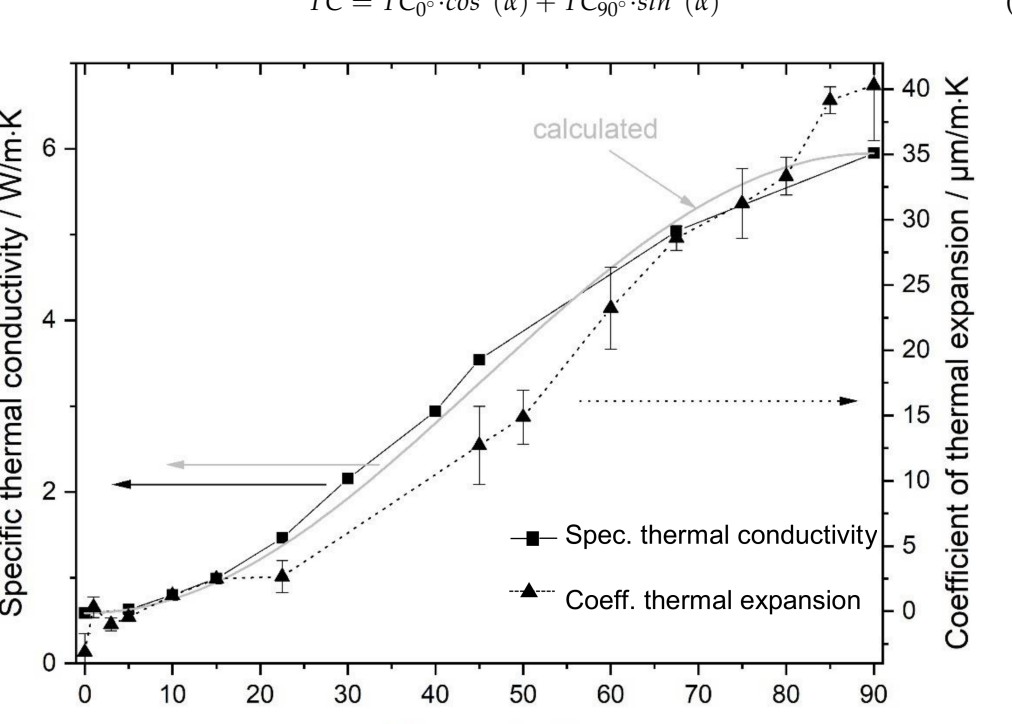

**Figure 5.** Thermal conductivity at 15 °C, additionally calculated according to Equation (1) and thermal expansion coefficient of CFRP with given out-of-plane fibre orientation.

The data shown in Figure 5 fit well with this simulation. The thermal expansion coefficient (Figure 5) is negative for an out-of-plane angle of 0°, which is typical for carbon fibres. At an out-of-plane fibre angle of 90°, 40 μm/m·K is measured, typical for the pure resin. For out-of-plane fibre angles in between, the thermal expansion coefficient shows a similar trend as the thermal conductivity. Further details of the thermal expansion and conductivity of 8552/IM7 with out-of-plane fibre angle are reported in [26].

### 4.2. Ablation Experiments

In principle, ablation properties can be tested by ASTM E285 [27]. Flat samples of ablative materials are treated with a flame of an oxyacetylene burner. Preceding experiments similar to ASTM E285 using an oxygen-propane flame turned out to be not suitable to systematically investigate ablation properties of the used epoxy-based CFRP. For the chosen experimental conditions with distances of 5 to 50 mm between welding gun and specimen, and test durations of 4 to 10 s, the material is heated up very rapidly and ignites. Figure 6 shows selected back side temperatures during the application of a propane flame on samples with no out-of-plane fibre orientation. After only 4 s of the flame impinging the sample from a distance of 5 mm, a rapid heat up on the samples back side is observed. This steep temperature rise continues after removing the pilot flame due to sustaining oxidative decomposition of the sample and persisting heat conduction to the back side. Maximum back side temperatures are reached in the range of 300 °C. For this experimental setup, it was not possible to reliably measure front side temperatures. However, they are assumed to be much higher than the necessary temperature for ignition of the material at 350 to 400 °C [25]. When applying the pilot flame for 10 s from a distance of 10 mm, the temperatures on the back side indicate an extensive decomposition and combustion of the matrix through the thickness of the specimen, as temperatures close to 600 °C are

reached and a sustaining flame is observed over 5 s after removing the pilot flame. The observed mass loss of 25% is typical for a combustion of 8552/IM7 samples tested by cone calorimetry [25]. Even for a larger distance of 50 mm between welding gun and specimen, the reached high back side temperatures indicate conditions of a rapid decomposition of the matrix in the CFRP material. For this experimental condition, the back side heats up a little slower and influences by formed delaminations are observed by an intermediate drop in the heating rate [42]. However, none of the 2 mm thick samples retain a significant residual strength after the conducted experiments.

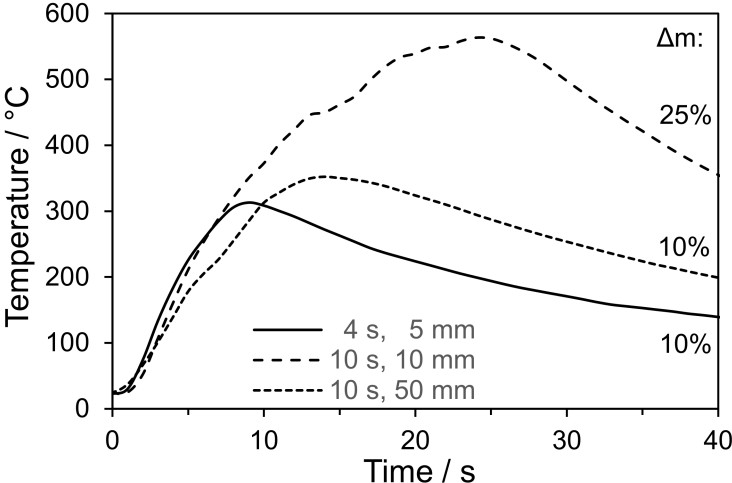

**Figure 6.** Temperature on the back side of samples with 0° fibre angle, during application of a propane flame for the given duration and distance between sample and welding gun, as well as corresponding mass loss (Δm) after the experiment.

This type of experiment is typically used to simulate ablative environments in rocket motors more than during aerodynamic heating [27]. Therefore, ablation properties were decided to be characterised by experiments using a heat-gut with 580 °C hot air. Carbon fibres withstand this temperature, as for their decomposition in air, typically a minimum threshold temperature of ~650 °C is necessary [43]. The polymeric components' epoxy resin and polyethersulfone degrade below 580 °C, with the epoxy resin being less thermally reistant than the polyethersulfone, as shown in the thermogravimetric analysis in Figure 7. However, no ignition of the samples is observed throughout the tests and test durations are long enough to differentiate the influence by out-of-plane fibre orientation.

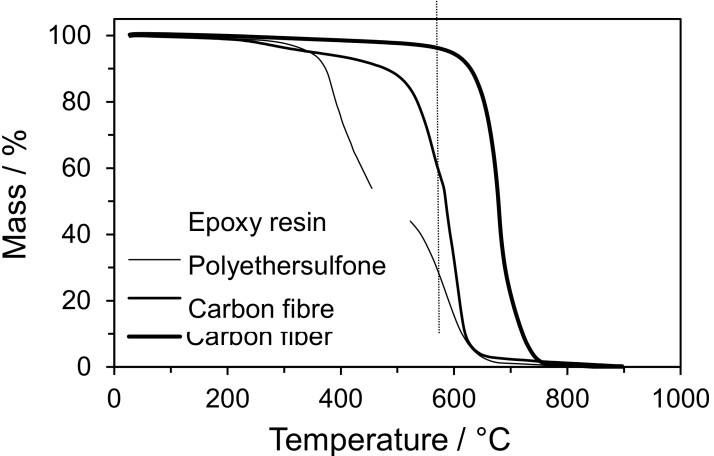

**Figure 7.** Thermogravimetric analysis of the separated components of the used CFRP. (Air temperature of 580 °C for ablation experiments is indicated.)

The recorded temperature rises on the back side of the samples with an out-of-plane fibre orientation are given in Figures 8 and 9, summarising maximum temperatures averaged for all conducted experiments with hot air after 60, 300 and 600 s. An equilibrium temperature is reached after ~25 s. For an out-of-plane fibre angle of 0°, a homogeneous temperature distribution (~240 °C) is observed at the samples' back side, as temperature curves at position 1 and 3 are nearly identical. For higher angles beginning from 3°, equilibrium back side temperatures are, in general, higher due to the increased thermal conductivity of the samples (see Figure 5) and especially higher at position 1 compared to position 3. For an angle of 15°, temperature difference between position 1 and 3 on the back side is ~70 °C (see also Figure 9). The preferred heat conduction along the fibres leads to an increasing temperature difference between position 1 and 3. At position 3, the reached temperatures are lower. As for high out-of-plane fibre angles, no fibres with a fibre end at the thermocouple position on the back side reached the front side. Such fibres only reached the lower edge of the sample (see inserted schemes in Figures 8 and 9). At position 1 and 2, these effects are less pronounced, and occurring temperatures better represent those of a large component. The observed temperature differences in the equilibrium state of the experiment prove that more heat is conducted to the samples´ back side for increasing out-of-plane angles.

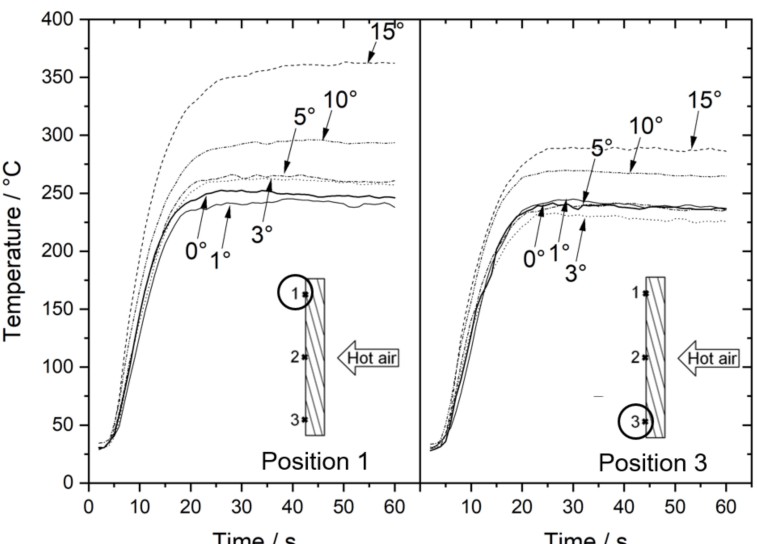

**Figure 8.** Temperature profiles on the back side of samples with different fibre angles during hot-air treatment: measured at position 1 (left); measured at position 3 (right).

The recorded equilibrium back side temperatures from 280 to 390 °C in Figure 9 are in a range of rapid matrix degradation of the 8552/IM7 composite [44]. High temperature gradients occur inside the samples with an assumed front side temperature close to the temperature of the impinging hot air. Therefore, a pronounced mass loss due to matrix decomposition is observed after the ablation experiment. However, the reached temperatures on the back side are lower compared to the impinging propane flame, and no ignition of the samples is observed. In irradiation experiments, temperatures of ~400 °C at the the front side were observed to be sufficient for ignition [25,44]. As temperatures are higher in the ablation experiment and no ignition occurs, it can be concluded that the rapid flow of hot air dilutes and removes formed pyrolysis gases, and critical concentrations therefore necessary for ignition are not achieved.

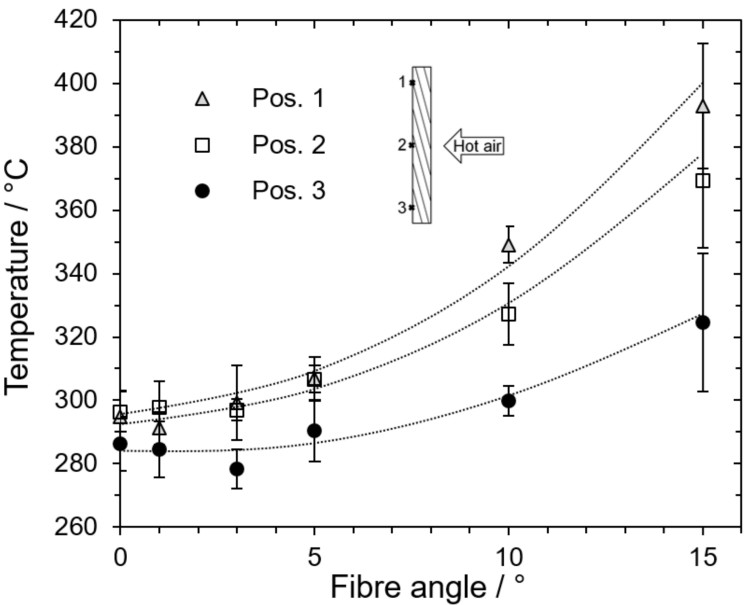

**Figure 9.** Reached equilibrium back side temperatures after ca. 30 s (see Figure 8).

*4.3. Material Characterisation after Thermal Impact*

To measure the effect of different fibre angles on thermal damage, first mass loss due to the decomposition of the polymer matrix is determined. The results are shown in Figure 10. Mass loss increases with time of thermal load (60, 300 and 600 s) but is more pronounced with an increasing fibre angle. As after ca. 25 s equilibrium back side temperatures are reached (see Figure 8), the increasing thermal decomposition of the resin matrix, in general, occurs with an increasing duration of thermal loading at certain temperature levels. However, when comparing samples with 0° and 15° out-of-plane angles, mass loss is 2 to 3 times higher for samples with 15°. According to the higher reached temperatures through the thickness of the panels with high out-of-plane fibre angles, extended matrix decomposition is responsible for the observed mass loss.

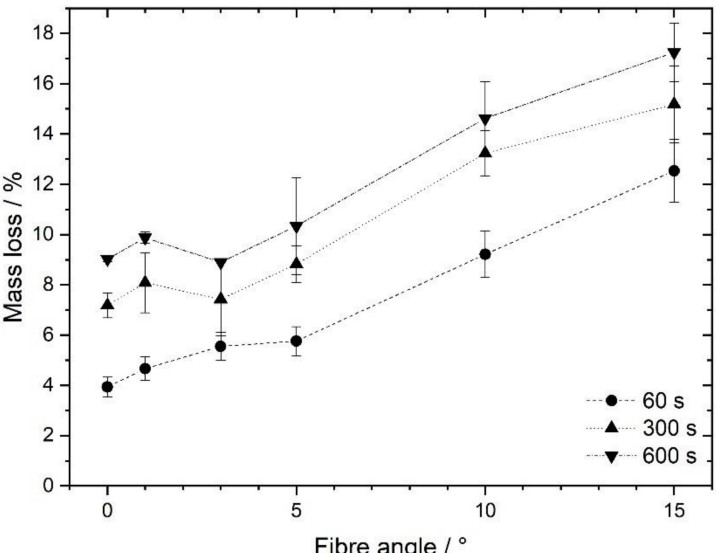

**Figure 10.** Mass loss of CFRP with an out-of-plane fibre orientation after different durations of hot-air treatment.

Increasing matrix decomposition with increasing out-of-plane fibre angles is also indicated by ultrasonic C-scans (Figure 11) and µCT cross-section images (Figure 12).

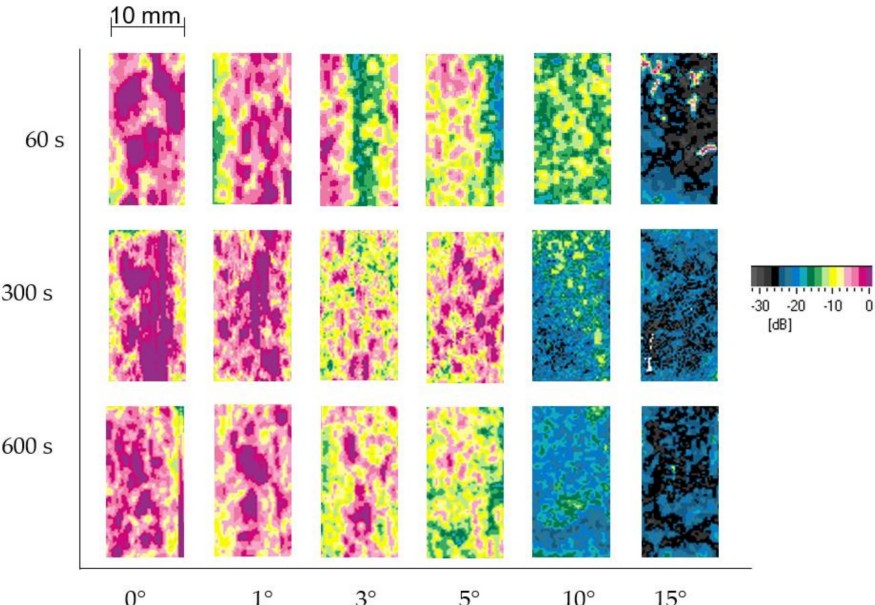

**Figure 11.** Ultrasonic C-scans from the back side of samples with various out-of-plane fibre angles after hot-air treatment for 60 to 600 s.

Ultrasonic C-scans from the back side of the samples indicate intact material on the back side for low out-of-plane fibre angles (0° and 1°). With increasing fibre angles, ultrasound is attenuated by deeper areas in the panel corresponding to a more pronounced decomposition of the matrix, which progresses from the front side to the back side of the samples with an increasing duration of hot-air treatment. Especially for the sample with an out-of-plane fibre orientation of 10° treated for 300 s, an inhomogeneous degradation can be observed on the back side. However, ultrasonic C-scans cannot significantly characterise matrix decomposition for this type of samples and are not ideal for measuring penetration depth compared to µCT. µCT images indicate that cracks and delaminations completely penetrate all investigated samples. However, pronounced matrix decomposition and areas of resin-depleted fibres are limited and increase with higher out-of-plane fibre angles. For low fibre angles up to 5°, matrix decomposition is observed to penetrate the material more homogeneously and less deeply. Matrix decomposition is favoured to occur along the fibres, as indicated in the xz cross-sections, especially for out-of-plane fibre angles of 10° and 15°. Areas at the edge of the samples, where fibres have no access to the hot surface, are less degraded. Here, adjacent fibres with access to the hot surface protect the matrix underneath, as heat is predominantly conducted along these fibres. Resin-depleted fibres act as a barrier for the heat transfer into the bulk material and for transport processes for decomposition products to the sample surface. However, these resin-depleted fibres with out-of-plane fibre angles between 3° and 10° are still locked in the resin matrix underneath, which can be seen by the additional white lines in Figure 12 representing the nominal out-of-plane angle.

In Figure 13, the maximum depths of resin-depleted areas are presented as being dependent on out-of-plane fibre angle after 60 and 300 s of thermal treatment. After 60 s, 0.73 mm of the material without out-of-plane fibre orientation is penetrated, corresponding to an average decomposition velocity of ~12 µm per second perpendicular to the fibres. With increasing out-of-plane fibre angles, the penetration depth increases. With a nearly linear increase in the penetrated depth for fibre angles up to 15°, an additional contribution to this velocity by the out-of-plane fibre angle of roughly 1.5 µm per second and angle degree is calculated (see linear regression in Figure 13: 0.088 mm/60 s = 1.47 µm/s). Therefore, decomposition velocity in depth direction is ~33 µm/s for the sample with an out-of-plane fibre orientation of 15°. For a hot-air treatment up to 300 s, a penetration depth of 1.1 mm is observed, and an average decomposition velocity of 3.6 µm per second

is calculated for the material with a 0° out-of-plane fibre orientation. An additional contribution to this velocity by the out-of-plane fibre angle is roughly 0.3 µm per second and angle degree (see Figure 13: 0.088 mm/300 s = 0.29 µm/s). Therefore, the sample with an out-of-plane fibre angle of 10° is already completely penetrated.

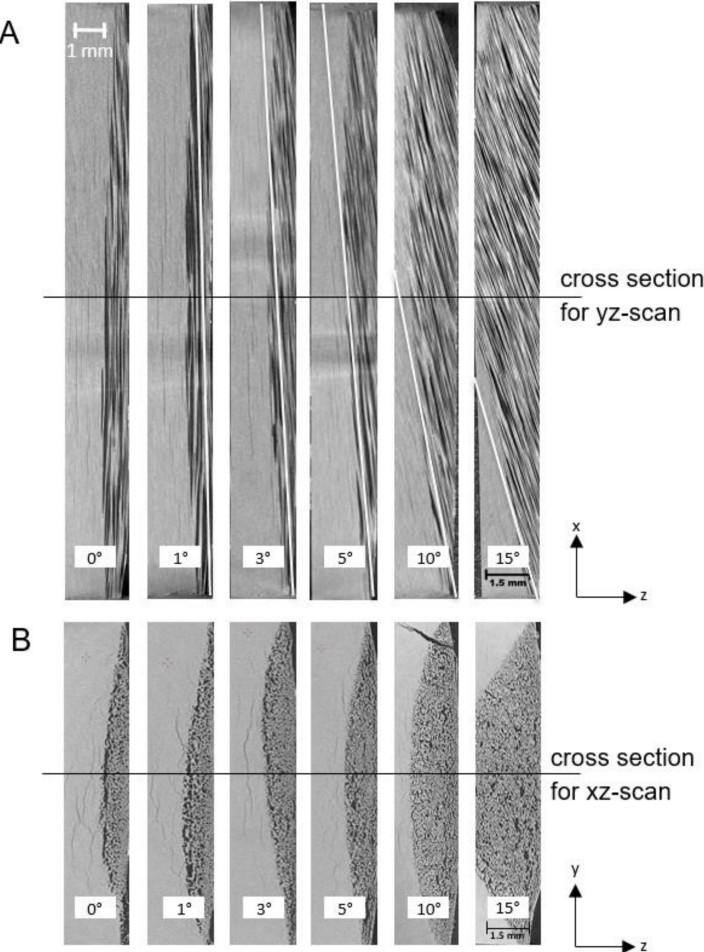

**Figure 12.** Microfocused computed X-ray tomography (µCT) analysis of CFRP samples with out-of-plane fibre orientation after 60 s of the hot-air treatment. (**A**): xz cross-section with indicated out-of-plane angle (white line); (**B**): yz cross-section.

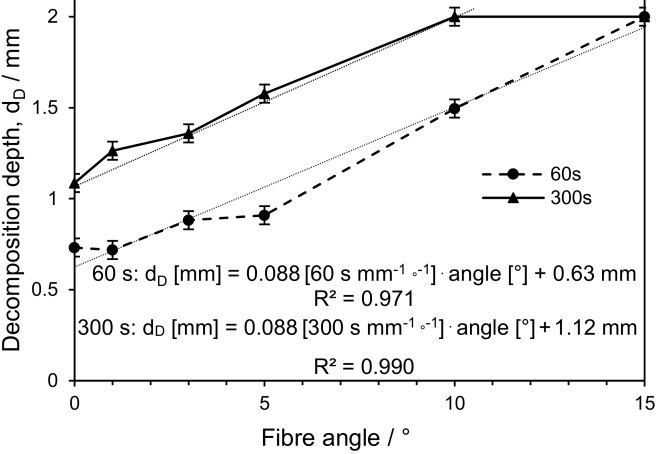

**Figure 13.** Maximum depth of matrix decomposition determined by µCT analysis (see Figure 9), and linear regressions for angle-dependent decomposition velocity (see text).

The decomposition of the polymer matrix leads to the formation of a liquid. This liquid is observed at the sample surface. In Figure 14, the surfaces of the samples with various out-of-plane fibre angles are shown after 60 s of hot-air impact. For an out-of-plane angle of 0°, loose fibres are observed, which are easily removed from the surface. For an angle of 1°, a rough surface can still be found, but it has less loose fibres. With an angle of 3°, the surface is formed like scales and no more loose fibres are found. Beginning from 5°, traces of a condensed, solid pyrolysate of the matrix are observed. Corresponding to the increasing mass loss and the reduced barrier effects for high out-of-plane fibre angles of 10° to 15°, the amount of condensed pyrolysate increases and a radial symmetric structure forms, which is caused by the impinging air.

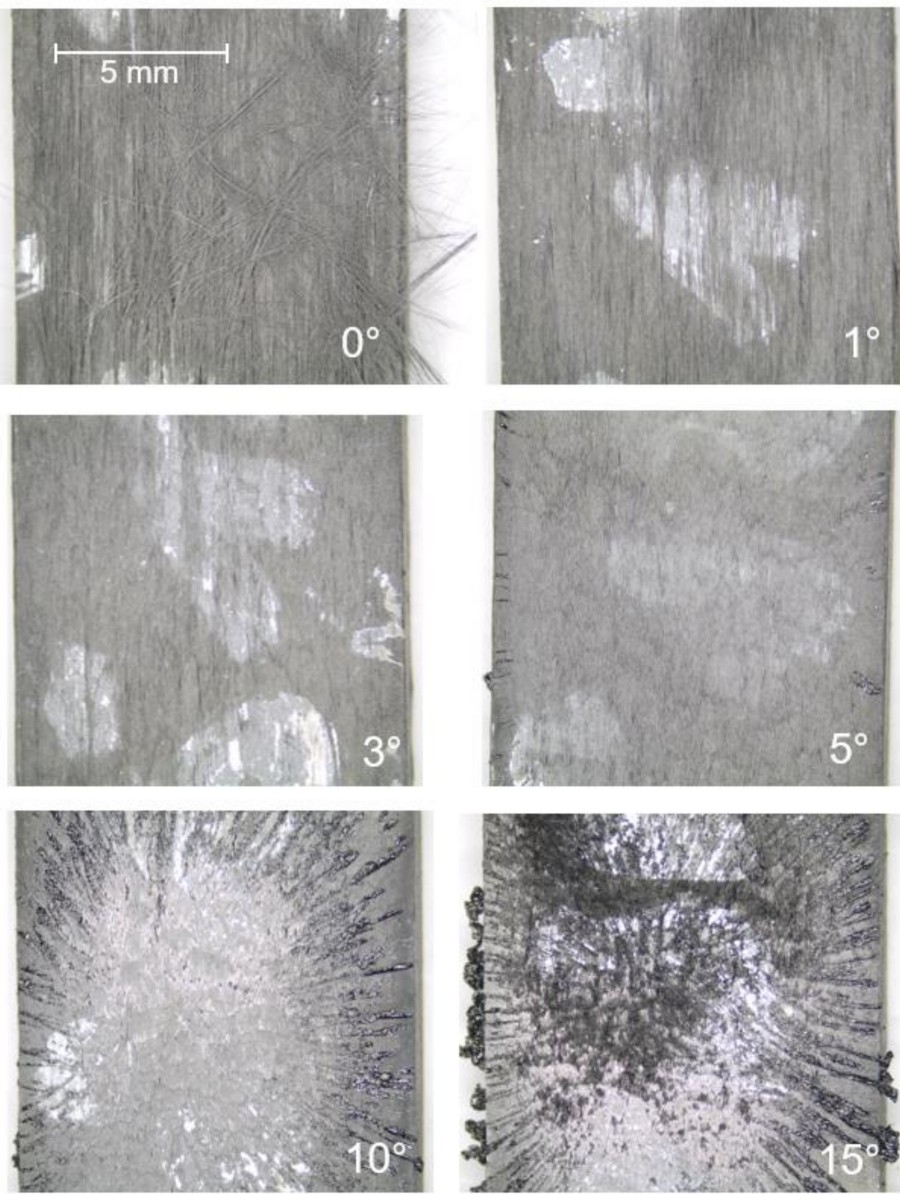

**Figure 14.** Images of the front side of samples with the given out-of-plane fibre angle after 60 s of the ablation experiment.

An SEM image of the surface of a 15° sample treated for 60 s (Figure 15) also shows the condensed pyrolysate and additionally proves that carbon fibres are not decomposed after thermal loading with hot air. The scale-shaped surface is formed by areas of fibre endings held together by residual polymer and surrounded by large cracks penetrating the material along the fibres. This observation is different from 8552/IM7 material in

irradiation experiments. In those experiments, the material ignites and combustion at higher temperatures forms loose char at the surface [25].

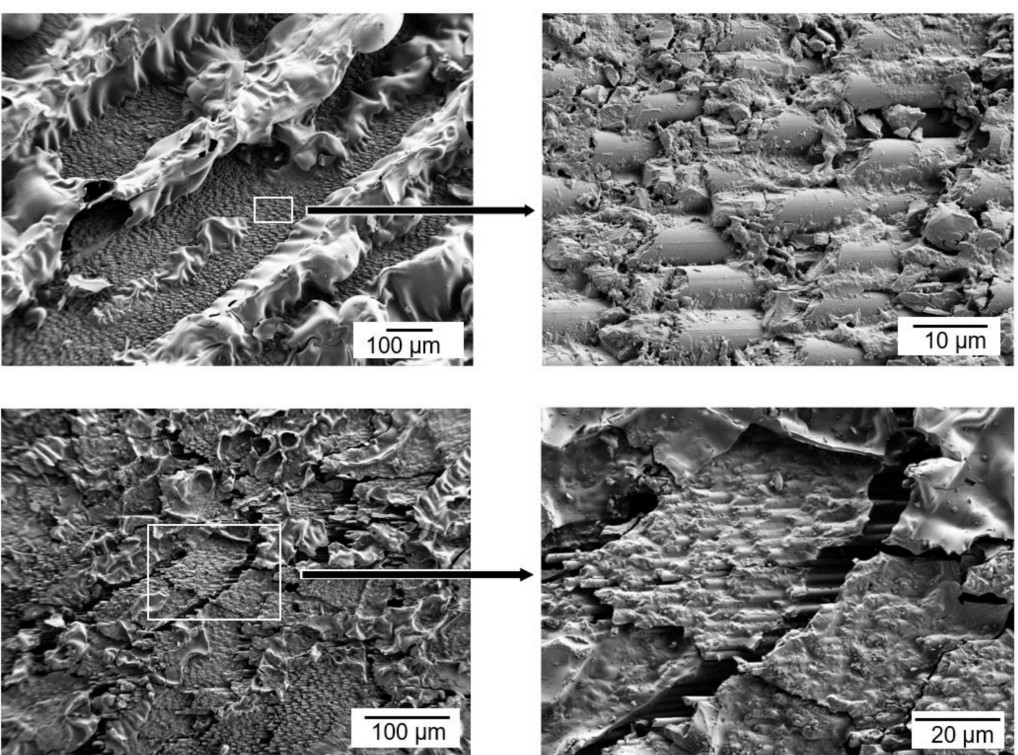

**Figure 15.** SEM photographs of the surface of a sample with an out-of-plane fibre angle of 15° after 60 s of hot-air treatment.

In summary, due to the increased out-of-plane fibre angle, more heat is conducted from the hot surface into the bulk of the material, and decomposition products are more easily transported to the surface. Barrier effects by the fibre plies are reduced. These effects increase the overall decomposition of the polymer matrix.

Moderate polymer degradation can be measured by FTIR spectroscopy on the back side of the samples. Figure 16 shows results for the samples with different fibre angles. The obtained spectra are typical for the 8552/IM7. A detailed assignment of bands is given in [29]. Characteristic bands at 1510 and 1486 cm$^{-1}$ are attributed to the epoxy resin and the polyethersulfone (PES), respectively [29]. A broad signal at 1650 cm$^{-1}$ originates from carbonyle species in oxidation products. With an increasing out-of-plane fibre angle, the band at 1510 cm$^{-1}$ loses intensity, which corresponds to a preferred degradation of the epoxy resin. As can also be seen in the thermogravimetric analysis, the epoxy resin in 8552/IM7 is less thermally resistant than the polyethersulfone. Therefore, a lower ratio of the intensities of the signals at 1510 and 1486 cm$^{-1}$ ($I_{1510\,\text{cm}}{}^{-1}/I_{1486\,\text{cm}}{}^{-1}$) corresponds to a more pronounced thermal degradation [26]. In Figure 17, this intensity ratio is given for the three positions on the back side of samples with different out-of-plane fibre angles after different durations of hot-air treatment. With increasing time, the observed intensity ratios are, in general, lower, indicating the proceeding thermal degradation of the polymer. For increasing fibre angles, a more pronounced polymer degradation occurs corresponding to the higher measured back side temperatures (see Figure 8). As expected, the thermal degradation of the polymer is only slightly depending on the fibre angle at position 3. At position 3, temperatures are lowest and least dependent on fibre angle. In contrast, at the other positions, matrix decomposition increases more strongly with an increasing fibre angle, as more heat is conducted to these areas on the back side of the samples. For example, after 60 s of thermal treatment, the samples with an out-of-plane angle beginning from 15° show a complete decomposition of the epoxy resin at the center of the samples´

back side (positions 1 and 2), whereas for the 10° sample, resin remains. After 300 s, the sample with a 10° out-of-plane fibre angle also shows a complete decomposition of the matrix at the center of the back side. These observations correspond to the maximum depths of resin-depleted areas traced by μCT (Figure 12).

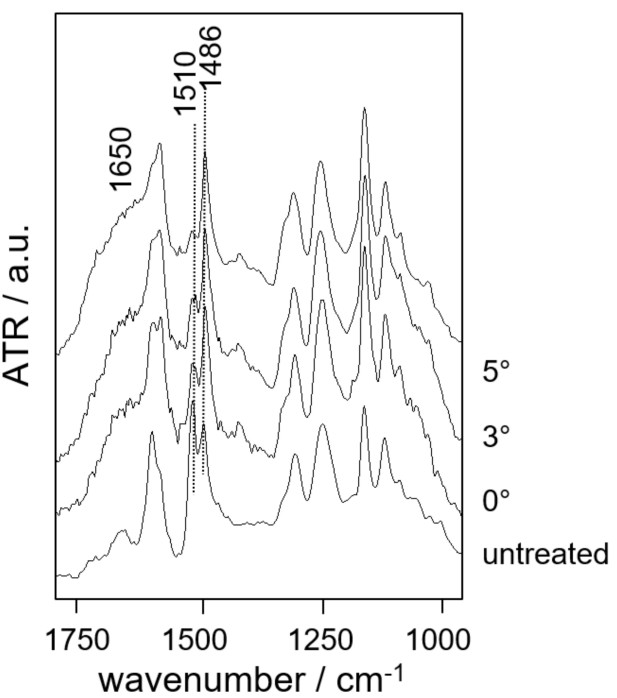

**Figure 16.** Infrared spectra (μ-ATR-FTIR) in the center of the samples' back side (position 2 in Figure 2) with a given out-of-plane fibre angle after 300 s of hot-air treatment.

*4.4. Residual Strength after Thermal Impact*

Thermally loaded samples are investigated for their residual short beam shear strength (SBSS). SBSS was chosen for mechanical testing, because the small area of thermal damage induced by the heat gun is best represented by the small samples (20 mm × 10 mm), and SBSS is less influenced by specimen geometry, for example, compared to the influence of the out-of-plane fibre angle on tensile strength (see above). Figure 18A presents the SBSS dependent on loading time. The drop in residual strength due to thermal loading is similar for samples with out-of-plane fibre angles between 0° and 3° within measurement tolerance. These samples lose one third of their initial SBSS after 600 s, whereas samples with out-of-plane fibre angles higher than 5° already completely degrade with respect to residual strength after 60 s of thermal treatment. An explicit comparison of residual strength is only possible for each fibre angle between various loading times (see Figure 18B), in order to not only consider the influence by the changing damage mechanism for various out-of-plane fibre angles. Within this comparison the relative decrease in SBSS is also similar up to an out-of-plane fibre angle of 3°. For angles higher than 3°, degradation is more pronounced with increasing angles. For 15°, residual strength is nearly completely lost after 60 s. In other words, loading time shows a significant influence on the residual SBSS of samples with low out-of-plane fibre angles, whereas for high fibre angles degradation is faster, and loading time is less decisive for the chosen conditions. In principle, the loss of SBSS can be correlated to matrix decomposition. Therefore, a first insight is given by the μCT analysis presented in Figures 12 and 13. More precisely, for the less decomposed matrix, the degradation of the polymer at the samples´ back side is correlated to residual SBSS in Figure 19 by means of infrared spectroscopy. In Figure 19, all samples are shown together with IR data representing position 1 on their back sides including all out-of-plane fibre angles and loading durations. Position 1 best represents back side conditions for a

larger component and typically stands for the most pronounced degradation compared to positions 2 and 3. With increasing matrix degradation, which corresponds to a decreasing intensity ratio, $I_{1510 \text{ cm}^{-1}}/I_{1486 \text{ cm}^{-1}}$, the residual SBSS, in principle, diminishes. This correlation comprises the increasing degradation of the matrix with the duration of thermal load and increasing out-of-plane fibre angle (see shaded symbols in Figure 19). Both duration and out-of-plane fibre angle significantly influence matrix degradation and residual SBSS. The correlation can be, in principle, used to non-destructively estimate the residual SBSS of a component with out-of-plane fibre angles. By measuring infrared spectra at the samples´ back side, an empirical prediction of mechanical performance can be achieved rapidly. However, standard deviations for the determined intensity ratios of the characteristic infrared bands are quite high, leading to not very precise estimations.

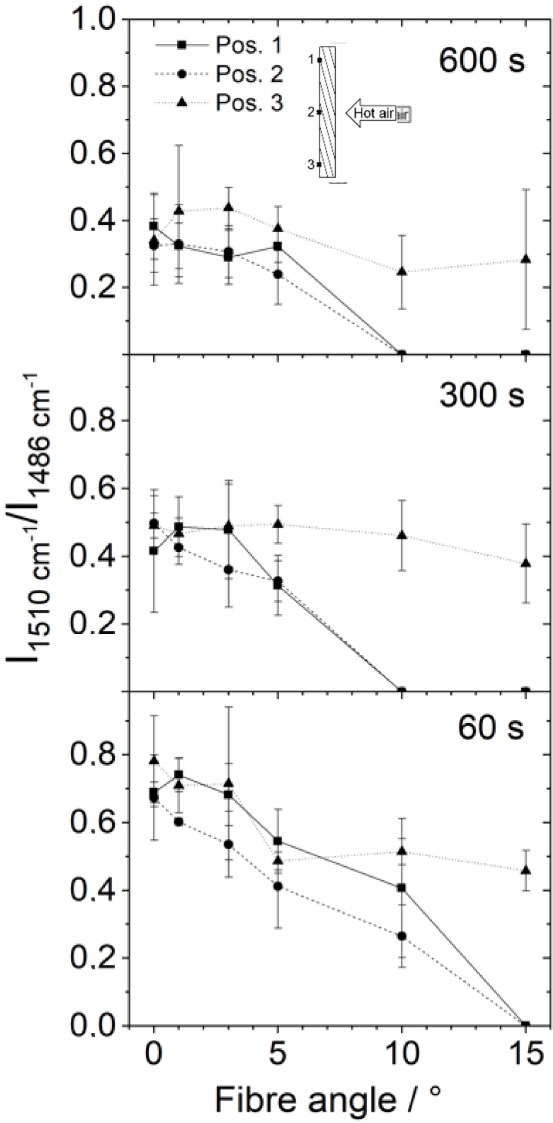

**Figure 17.** Intensity ratio of IR bands at 1510 cm$^{-1}$ (EP) and 1486 cm$^{-1}$ (PES) recorded at different positions on the back side of the samples after hot-air treatment for different times.

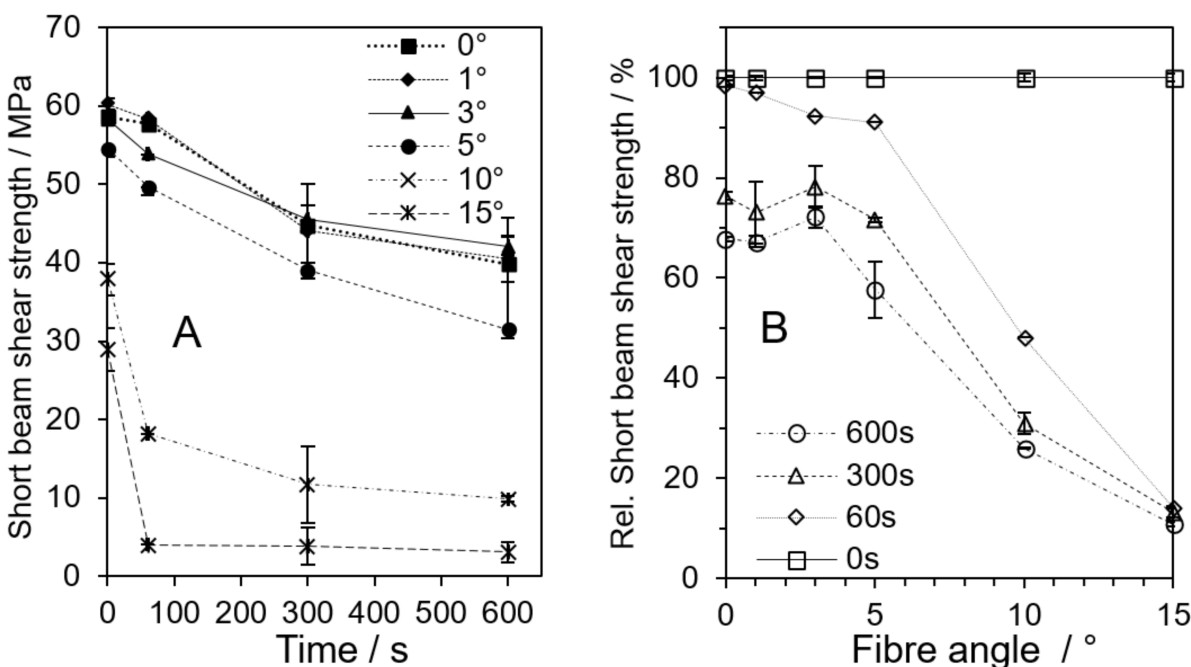

**Figure 18.** Short beam shear strength of samples after hot-air treatment. (**A**): dependent on loading time; (**B**): each sample related to the corresponding sample with the same out-of-plane fibre angle but without thermal treatment (0 s).

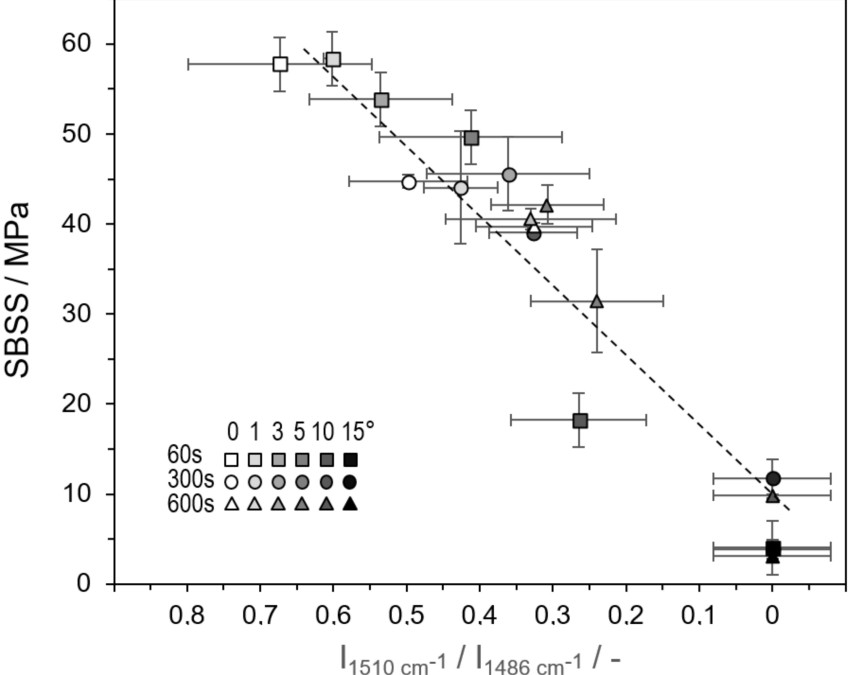

**Figure 19.** Correlation of short beam shear strength (SBSS) and matrix degradation by means of the intensity ratio of IR bands at 1510 cm$^{-1}$ (EP) and 1486 cm$^{-1}$ (PES) at position 1 on the back side of samples for all out-of-plane fibre angles and test durations. (The dashed line is introduced for clarity reasons only.)

## 5. Conclusions

This work assesses the potential use of CFRP with an out-of-plane fibre orientation for structural components under ablative conditions. Therefore, samples of an epoxy-based CFRP with low out-of-plane fibres are examined using a hot-air blower. Achieved temperatures are high enough to decompose the matrix but not the fibres. Up to an out-

of-plane fibre angle of ~3°, a slight reduction in initial mechanical performance seems negligible for applications in structural components. The thermal conductivity through thickness of intact CFRP rises with an increasing out-of-plane fibre angle and penetration depth, as barrier effects for the migration of decomposed material by the fibres are reduced. However, resin-depleted fibres are fixed by the partially intact matrix material underneath, keeping them in position in a real ablative application.

Occurring temperature and measured residual strength can be well correlated to the observed matrix degradation characterised by infrared spectroscopy on the samples´ back side. Therefore, infrared spectroscopy might also be used to rapidly predict residual strength for CFRP samples with an out-of-plane fibre orientation.

The conducted experiments in principle describe a successful way to apply out-of-plane fibre orientation for ablative materials. An out-of-plane fibre orientation of ~3° is identified as a good compromise between loss of strength and thermal conductivity as well as matrix degradation. It is expected that the observed effects are more pronounced in samples thicker than the used 2 mm samples. Additionally, matrices as well as fibres with higher thermal resistance are supposed to improve ablation performance. Matrices containing high-temperature-resistant polymers and reinforcements with non-combustible fibres seem especially interesting. A further way to use fibres with an out-of-plane orientation in ablative processes is to fix them in thermally resistant substrates and apply ablative layers, which are penetrated by the fibres. The fibres prevent the ablative layers from being easily removed. Matrix-depleted fibres with out-of-plane angles at the surface may additionally positively influence the laminar flow of air in high-speed applications.

Samples used for this investigation were complex to manufacture. It is far easier to apply winding techniques, for example, with a slight lateral offset of the applied prepreg layers to prepare CFRP pipes with an out-of-plane fibre orientation [28]. Obtained results may also apply for composites with partial out-of-plane fibre reinforcement, such as fabrics or braided structures.

The conducted experiments take a longer time compared to typical hypersonic applications of one-way parts with only several seconds. Therefore, the performance of fibre-reinforced composites with an out-of-plane fibre orientation might be sufficient to warrant their use as a structural lightweight material with improved ablation behaviour in the future.

**Author Contributions:** Conceptualization, S.E.; methodology, S.E., T.J.S. Schuster; validation, S.E., T.J.S. Schuster; formal analysis, S.E., T.J.S. Schuster; investigation, S.E., T.J.S. Schuster; resources, S.E., T.J.S. Schuster.; data curation, S.E., T.J.S. Schuster; writing—original draft preparation, S.E., T.J.S. Schuster; writing—review and editing, S.E.; visualization, S.E., T.J.S. Schuster; All authors have read and agreed to the published version of the manuscript.

**Funding:** This research received no external funding.

**Institutional Review Board Statement:** Not applicable.

**Data Availability Statement:** Data are available by request from S.E.

**Conflicts of Interest:** The authors declare no conflict of interest.

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
