# Peer review of "Thermal and Principal Ablation Properties of Carbon-Fibre-Reinforced Polymers with Out-of-Plane Fibre Orientation"

_carbon_

Round 1
Reviewer 1 Report
This paper theoretically studied the thermal and principle ablation properties of carbon fibre reinforced polymers with out-of-plane fibre orientation. The paper presented comprehensive results and could be considered for publication if minor revisions are made to address the following raised comments:
- What is the merit of the current work compared to previous studies? The authors may need to highlight the novelty of the work after discussing existing work.
- ASTM e285 was adopted for the test. It is recommended to mention the key points and main purpose of this standard to help readers to better understand the test.
- The quality of the figures should be improved. There are several figures having words/number missing. The authors may need to do careful proofreading before re-submission.
- The conclusion, which includes repeated information, is too long. It is recommended to shorten this section to make it more concise.
Author Response
see attached word doc.

Reviewer 2 Report
The work described in the paper is interesting but there are some deficiencies before acceptance, few points are –
- In introduction, please try to avoid citing bulk references such as [2-8], [11-14] and so on. Please describe them to individual references. Please check this point in whole manuscript. In addition, please refer 2-3 papers from carbon-MDPI journal in field of present paper and describe the advancements of the present work from the reported work.
- In Figure 3, authors claim that the mechanical properties decreases with an increase in fibre angle but did not describe why satisfactorily in interpretation section? How authors determine compressive strength is not detailed? It would be interesting if authors can show stress-strain curves of the composites as a function of fibre angle. Moreover, in Figure 4, the scale bar is not visible.
- In Figure 5, authors shows that the thermal properties increases with increasing fibre angle. Is it due to improved inter-fibre networking with increased fibre angle and formation of long range continuous fibre networks? Please comment on this aspect?
- In Figure 7, it is not clear which TGA curve belong to which sample? Please cross check? In Figure 8, what do author mean with positions? Are they spots at different locations? Please clear this point? Moreover, author did not talk about position 2? In Figure 9 also, it is not clear which curve belong to which point or sample?
- What do authors want to communicate from Figure 12? It is not clear from the study and its discussion? Please elaborate on this point? Moreover, in Figure 15, the sale bar is not visible?
Author Response
see attached word doc.

Reviewer 3 Report
Authors described a interesting study on carbon fibers composites. Nontheless, it is not so innovative and it presents some issues listed as follow.
Do not use any acronyms in the abstract.
What do authors mean with "this could e.g. be achieve" at line 40?
In figures 14 and 15 scales must be added.
Ire spectra are very poorly reported and discussed. A detailed description of the spectra must be included. Authors cannot just commented two of a multitude peaks. Bands must be identified by using the appropriate nomenclature with Greek letters.
Nevertheless, the real weak point of the work is the total absence of any statistical treatment of the data reported. How could authors say that one data is different form another one or enlighten a trend?
A deep reconsideration of the data set provided should be included by using two-way ANOVA test and where it is necessary a t-test.
Considering the issues raised above, I cannot endorse the publication of this paper prior without major revisions.
Author Response
see attached word doc.

Round 2
Reviewer 2 Report
Accept in present form
Reviewer 3 Report
Questions were answered.